# Recent Advancements in Non-Invasive Vaccination Strategies

**DOI:** 10.3390/vaccines13090978

**Published:** 2025-09-16

**Authors:** Mahek Gulani, Tanisha Arte, Amarae Ferguson, Dedeepya Pasupuleti, Emmanuel Adediran, Yash Harsoda, Andrew Nicolas McCommon, Rikhav Gala, Martin J. D’Souza

**Affiliations:** 1Vaccine Nanotechnology Laboratory, College of Pharmacy, Center for Drug Delivery Research, Mercer University, Atlanta, GA 30341, USA; mahekanil.gulani@live.mercer.edu (M.G.); tanisha.manoj.arte@live.mercer.edu (T.A.); amarae.ferguson@live.mercer.edu (A.F.); emmanuel.adediran@live.mercer.edu (E.A.); yashkumar.pankajbhai.harsoda@live.mercer.edu (Y.H.); 2Department of Pharmaceutical Sciences, Larkin College of Pharmacy, 18301 North Miami Ave., Miami, FL 33169, USA; dpasupuleti@larkin.edu; 3Macon Campus, Mercer University of Liberal Arts and Sciences, Macon, GA 31207, USA; andrew.nicolas.mccommon@live.mercer.edu; 4Drug Product Design and Development, Biotherapeutic Pharmaceutical Sciences (BTx), Pfizer Inc., Andover, MA 01810, USA; rikhav.gala@pfizer.com

**Keywords:** noninvasive, vaccine, vaccination, microneedle vaccine, intranasal vaccine, sublingual vaccine, vaginal vaccine, immunization

## Abstract

Vaccines remain one of the most powerful tools in modern medicine, having revolutionized public health by preventing millions of deaths and controlling the spread of infectious diseases worldwide. However, conventional needle-based vaccines face several limitations, including pain and discomfort, the need for cold-chain infrastructure, reliance on trained healthcare personnel, risk of cross-contamination, and limited accessibility in low-resource settings. These challenges have spurred the development of non-invasive vaccination approaches that promise safer, more accessible, and patient-friendly immunization. Non-invasive immunizations not only eliminate the need for needles but may also enhance compliance and enable mucosal immune responses. To harness the full potential of these innovative delivery routes, a comprehensive understanding of their formulation strategies and mechanism of action is essential. This review aims to comprehensively discuss recent advancements in oral, intranasal, microneedle, buccal, sublingual, and vaginal vaccinations and highlight their underlying immunological mechanisms, formulation strategies in preclinical studies, examples of marketed products, and ongoing clinical trials.

## 1. Introduction

Many infectious diseases such as malaria, meningitis, diphtheria, hepatitis B, tetanus, polio, measles, mumps, and rubella, are now well managed in developed countries through vaccination programs and robust healthcare systems, there remains a critical need to enhance global healthcare systems as multiple infectious disease outbreaks worldwide such as respiratory tract infections, HIV/AIDS, malaria, tuberculosis, dengue, and Influenza, take place yearly. They account for more than fifty percent of overall mortality. Techniques are urgently required to develop specific protective vaccines, which are currently not available for some diseases. World Health Organization vaccination efforts have decreased death rates, but advances in vaccine administration and storage are still needed. Temperature fluctuations can damage vaccine vials, requiring them to be disposed [1].

Vaccine delivery approaches can be categorized into invasive, minimally invasive, and non-invasive techniques. Traditional immunization predominantly employs invasive administration to ensure high bioavailability. Vaccination is now referred to as ‘getting one’s shot’ among people and is associated with pain and discomfort [2]. Conventional vaccines such as diphtheria–tetanus–pertussis (DTP) and their combination formulations (e.g., DTP-HepB-Hib, DTP-HepB-Hib-IPV) represent some of the most widely used and successful immunization strategies globally. These injectable vaccines have played a pivotal role in reducing childhood morbidity and mortality in both developed and developing countries, and they continue to serve as a cornerstone of routine immunization programs recommended by the WHO and national agencies. Despite their effectiveness, these approaches face practical challenges that can hinder their widespread use. The reliance on liquid formulations strictly requires cold-chain system, for storage and transport, which most vaccines, mRNA-based vaccines being particularly sensitive, require as they are highly vulnerable to temperature fluctuations which can compromise vaccine integrity. For example, for the COVID-19 vaccine, across various manufacturers including Pfizer–BioNTech, Oxford-AstraZeneca, and ModernaTX, the shelf life reduces with increase in temperature. Secondly, needle-based vaccination poses risks such as accidental needlestick injuries and the potential spread of bloodborne infections at the injection site, including hepatitis B, hepatitis C, and HIV. Technical issues such as differences in adipose tissue thickness among individuals require optimizing needle length based on factors like body weight and sex. Ensuring correct vaccine delivery often demands trained professionals and appropriate infrastructure, which can be challenging in remote areas and resource-limited settings. Thirdly, invasive immunization mainly induces systemic immunity in contrast to natural infections which induce systemic as well as mucosal immunity. Most infectious agents enter the body via the mucosa, which forms the barrier between the interior of the body and its external surroundings [3]. The inadequate mucosal antibodies can prevent the body from eliminating the pathogens at the start of disease which may make the disease more severe. To overcome this, noninvasive vaccination can induce systemic and mucosal antibodies effectively avoiding pathogens from causing serious illness [1].

Vaccine delivery in the current era is predominantly based on an invasive route of administration and has showed promising results when administered accurately. As a result of the above-mentioned limitations, the need for new vaccine options is increasing, leading to active research into minimally invasive and non-invasive vaccines as promising alternatives. This review focuses on latest advancements in the rapidly evolving field of non-invasive vaccines—oral, intranasal, microneedle, buccal, sublingual, and vaginal vaccinations—that have the potential to elevate immunization practices, aiming to shed light on the advantages, mechanism of action, and the currently approved vaccinations or in clinical trial stages.

## 2. Oral Vaccines

### 2.1. Early Development of Oral Vaccines

Introduced in the 1960s, oral polio vaccine enabled large-scale immunization campaigns that significantly reduced global polio incidence. Traditional oral vaccines face several limitations such as antigen degradation in the harsh environment of the gastrointestinal tract including acidic pH and digestive enzymes which can degrade vaccine antigens before they reach immune-inductive sites, necessitating higher antigen doses for efficacy [4,5].The efficiency of antigen presentation in the gut-associated lymphoid tissue (GALT) is limited, sometimes resulting in weaker or inconsistent immune responses compared to injectable vaccines [6]. Some oral vaccines, like those for cholera, provide only temporary protection and require booster doses to maintain immunity [7,8].

### 2.2. The New Wave of Oral Vaccine Research

#### 2.2.1. Nanoparticle and Microparticle Delivery Systems

Nanoparticulate platforms work by encapsulating vaccine antigens within protective carriers, which shield them from the harsh acidic environment and digestive enzymes of the gastrointestinal tract. This protection ensures that antigens remain intact until they reach immune-inductive sites, thereby preserving their immunogenicity [9] (Figure 1).

A major advantage of these systems is their ability to facilitate targeted delivery. By engineering nanoparticles and microparticles to interact specifically with cells in the gut, such as M cells and dendritic cells, researchers can optimize the uptake of antigens and enhance both mucosal and systemic immune responses [10]. Additionally, these carriers can be designed for controlled release, which can result in prolonged immune activation and potentially reduce the need for booster doses. Some nanoparticle formulations also possess inherent adjuvant properties, further stimulating the immune system without the need for additional components [11,12]. Large-scale, consistent manufacturing of nanoparticles with uniform size and antigen loading is demanding, and regulatory pathways for these novel delivery systems are still being established. Preclinical study for COVID-19 utilize polymeric nanoparticles and encapsulated spike protein or RBD [13]. Another preclinical study for Hepatitis B utilize albumin–chitosan microparticles encapsulating plasmid DNA or HBsAg for oral use [14]. A study for oral cholera vaccine used poly(lactide) microparticles encapsulating CTB, inactivated vibrios, or carrier protein [12].

#### 2.2.2. Recombinant Microbial Vectors in Oral Vaccines

Recombinant microbial vectors, including genetically engineered bacteria and yeast, are increasingly utilized as vehicles for oral vaccine delivery due to their inherent resistance to the gastrointestinal environment and capacity to induce robust immune responses. Among these, *Bacillus subtilis* spores exhibit exceptional stability, are recognized as safe for consumption, and can act as natural adjuvants [15]. Additionally, the spores can display antigens on their surface or within vegetative cells, and their natural resistance to heat, pH, and desiccation makes them ideal for oral administration.

Lactic acid bacteria (LAB), such as *Lactobacillus plantarum* and *Lactococcus lactis*, are acid-resistant and capable of efficiently targeting Peyer’s patches in the intestine, making them effective for delivering antigens to mucosal immune tissues and stimulating strong secretory IgA and systemic antibody production [16].

Yeast vectors, notably *Saccharomyces cerevisiae* and *Pichia pastoris*, provide added benefits through their robust cell walls, which protect encapsulated antigens from gastrointestinal degradation, and their flexibility for genetic engineering [17]. These vectors not only shield antigens, enhancing bioavailability, but also stimulate both mucosal and systemic immune responses, sometimes allowing for transient gut colonization to prolong antigen exposure. Efficient and consistent antigen expression, the need for high antigen doses or repeated administration, regulatory concerns regarding safety and genetic stability, and the possibility of reduced efficacy with repeated use due to host immune responses are ongoing hurdles. *Bacillus subtilis* spores are safe, spore-forming, and scalable vectors that can withstand the harsh gastrointestinal (GI) environment, present antigens, and trigger both innate and adaptive immune responses. They have been explored for vaccines against tetanus, rabies, and rotavirus, though they require repeated dosing and pose regulatory and genetic safety challenges. Lactic acid bacteria (LAB), such as *L. plantarum* and *L. lactis*, are acid-resistant and can target Peyer’s patches, where they colonize the gut, present antigens to mucosal immune cells, and induce secretory IgA and systemic IgG responses. They have been investigated for vaccines against enterotoxigenic *E. coli*, COVID-19, and Lyme disease, but face issues of variable immunogenicity, production difficulties, and unpredictable host immune responses. Yeast-based vectors, including *S. cerevisiae* and *P. pastoris*, have robust cell walls and are highly engineerable, enabling them to protect antigens, promote immune uptake, and induce both humoral and cellular immunity. They have been applied in hepatitis B and influenza vaccines and evaluated in various preclinical and clinical studies; however, achieving consistent antigen expression and meeting regulatory requirements remain challenges.

*Bacillus subtilis*-based vaccines have been evaluated in several preclinical studies. For tetanus [18], recombinant *B. subtilis* spores or vegetative cells expressing the tetanus toxin fragment C (TTFC), sometimes combined with *E. coli* mLT as an adjuvant, have been tested in mice and pigs. Oral rabies vaccine candidates [19] such as CotG-E-G and CotG-C-G, displaying rabies virus glycoprotein, showed promise in mice. Similarly [20], recombinant *B. subtilis* expressing rotavirus proteins (VP8* or VP6), often delivered as spores or vegetative cells and sometimes paired with mucosal adjuvants like cholera toxin or *E. coli* mLT, were evaluated in mice. Lactic acid bacteria (LAB) have also been explored [21]: *Lactococcus lactis* NZ3900 and *Lactobacillus casei* were engineered against enterotoxigenic *E. coli* (ETEC), expressing mutant heat-labile toxin subunits (LTA, LTB) and F4 fimbriae (FaeG), and tested in piglets and mice. LAB strains including *L. lactis* IL1403, NZ3900, and *Lactiplantibacillus plantarum* have been used to deliver SARS-CoV-2 spike or receptor-binding domain (RBD) antigens in mouse COVID-19 models [22]. For Lyme disease [23], *L. plantarum* was engineered to present *Borrelia burgdorferi* surface antigens such as OspA and was studied in mice. Yeast-based vectors have reached both preclinical and clinical testing: recombinant *Saccharomyces cerevisiae* expressing hepatitis B surface antigen (HBsAg) [24] underlies licensed vaccines like Engerix-B and Recombivax HB, as well as oral prototypes; and yeast-displayed influenza vaccines [25], including H5N1 hemagglutinin and recombinant virus-like particle platforms, have been investigated preclinically.

#### 2.2.3. Mucoadhesive and Mucus-Penetrating Formulations in Oral Vaccine Delivery

Mucoadhesive and mucus-penetrating formulations are designed to address the key challenge of ensuring that vaccine antigens remain at mucosal surfaces long enough to be effectively absorbed and presented to the immune system. Mucoadhesive delivery systems leverage materials such as chitosan, alginate, carbopol, gelatin, and other natural or synthetic polymers that can adhere to the mucus layer lining the oral, buccal, or intestinal mucosa. This adhesion prolongs the retention of antigens at their site of administration, increases the chances of uptake by mucosal immune cells like M cells and dendritic cells, and protects the antigens from both rapid mucosal turnover and enzymatic degradation. The result is sustained antigen release and a greater opportunity for immune activation [26,27] (Figure 2).

In addition to mucoadhesion, newer mucus-penetrating technologies are being developed to allow vaccine carriers to traverse the thick mucus barrier and reach immune-inductive tissues beneath, such as the epithelial layer of the gut or oral mucosa. These formulations often utilize nanoparticles, surface modifications like PEGylation, or specialized coatings that reduce binding to mucin fibers, enabling deeper diffusion through the mucus network. This approach enhances delivery directly to key immune cells for more robust immune responses [28].

Innovations in this area include thermoresponsive gels, microneedle systems for oral mucosa, composite films and particles, and materials that respond to local environmental cues for optimal release and targeting [29,30,31,32,33] Several preclinical and early clinical studies have explored mucoadhesive vaccine delivery systems for respiratory viruses such as COVID-19 and influenza. For COVID-19, mucoadhesive buccal films incorporating platforms like mRNA/LNPs, DNA, and viral vectors have been formulated using polymers such as polyvinyl alcohol (PVA), hydroxypropyl methylcellulose (HPMC), and Carbopol [34]. Similarly, influenza vaccines using mucoadhesive films and nanoparticles have utilized influenza hemagglutinin (HA) proteins or synthetic peptides with various adjuvants, employing polymers like chitosan, poly(anhydride), PVA, and Carbopol [34].

#### 2.2.4. Bacterium-like Particles (BLPs) and Lipid-Coated Systems in Oral Vaccines 

Bacterium-like particles (BLPs) are non-living, hollow peptidoglycan skeletons derived from lactic acid bacteria, created by removing proteins and nucleic acids through heat and acid treatment [35]. These particles retain the immunostimulatory properties of the bacterial cell wall and serve as safe, stable platforms for antigen delivery. BLPs can be engineered to display various vaccine antigens on their surface using protein anchor technology, which allows for efficient antigen presentation [35]. One of the main advantages of BLPs is their high safety and stability because they are devoid of genetic material, there is minimal risk of infection or genetic transfer, and their structure is highly resilient under environmental stress. Additionally, the peptidoglycan structure of BLPs acts as a natural adjuvant, activating innate immune responses via toll-like receptor 2 (TLR2) and enhancing both mucosal and systemic immunity [36]. The flexibility of protein anchor systems further enables the attachment of diverse antigens, making BLPs a versatile platform for various vaccine applications.

Despite these advantages, the harsh gastrointestinal environment, characterized by acidic pH and digestive enzymes, can degrade surface-displayed antigens, thereby reducing vaccine efficacy [37]. To address this, recent research has focused on encapsulating BLPs within lipid membranes, which has significantly improved the prospects for oral BLP vaccines. The lipid coating provides enhanced protection for surface antigens, shielding them from degradation in the gastrointestinal tract and preserving their immunogenicity. Lipid-coated BLPs have also demonstrated improved absorption in intestinal Peyer’s patches, which are critical sites for initiating mucosal immune responses. Notably, recent studies have achieved up to 99% encapsulation efficiency with these systems, ensuring that most antigens remain protected during transit [35].

The impact of these innovations is evident in preclinical studies, where a single oral immunization with lipid-coated BLP vaccines has induced rapid, strong, and cross-protective immunity, with up to 100% protection in animal challenge models [35]. Mechanistically, lipid-coated BLPs act as both antigen carriers and adjuvants. The lipid membrane protects the antigen as it passes through the stomach and upper intestine, and upon reaching the lower intestine, facilitates absorption in Peyer’s patches. Here, BLPs interact with immune cells to stimulate both innate and adaptive immunity, resulting in enhanced overall immunogenicity of oral vaccines [38].

#### 2.2.5. Plant-Based Oral Vaccines 

Plant-based oral vaccines utilize genetically engineered plants to produce vaccine antigens, which are then delivered by consuming the edible parts of the plant, such as leaves, fruits, or seeds [39]. This method leverages the natural bioencapsulation provided by plant cells, which helps protect antigens from degradation in the gastrointestinal tract and facilitates their uptake by the immune system [40]. This bioencapsulation enhances the safety and stability of plant-based vaccines, and the approach is highly scalable and cost-effective, as plants can be cultivated on a large scale at low cost. Furthermore, edible plant tissues can be consumed directly, eliminating the need for antigen purification, cold-chain storage, or trained healthcare personnel, and reducing the risk of contamination associated with animal-derived products [39,41]. Examples include transgenic potatoes, tomatoes, and bananas expressing antigens for diseases such as hepatitis B, cholera, and Norwalk virus, as well as rice and maize used for vaccines against enterotoxigenic *E. coli* and rotavirus [42,43,44,45]. In parallel, bacterial-like particles (BLPs) derived from lactic acid bacteria have been studied as oral vaccine platforms. For enteric bacterial infections [35], candidates such as COB17 and LM@COB17 used lipid-coated BLPs conjugated with model protein antigens, tested in preclinical settings without additional adjuvants. Similarly, BLPs have been engineered to display HIV-1 gp120 Env trimer antigens, representing a preclinical oral vaccine approach against HIV-1.

## 3. Inhalation Vaccines

### 3.1. Intranasal Vaccines

The first intranasal vaccine formulated against influenza and licensed in 2003 and approved in the USA market is the Flumist. It is a live attenuated influenza vaccine (LAIV) which has quadrivalent influenza virus subtypes A and B strains, approved for ages 2 to 49. Fluenz is the marketed name of the similar composition of the vaccine that is approved in the European market [46,47].

Intranasal delivery is a patient-friendly route causing minimal discomfort to children, elderly, and people with needle phobia. In addition, self-administration is possible with this route and that leads to easy vaccination and faster immunization for a large population, especially in the case of a pandemic. By targeting the highly vascularized and immunologically active nasal mucosa, this non-invasive approach enables rapid systemic absorption and efficient stimulation of mucosal immunity [48,49].

#### 3.1.1. Mechanism of Action of Intranasal Administration

Mucosal sites are major attractions for respiratory viruses, causing the spread of the infection [50]. Mucosal lining comprises specialized mucosa-associated lymphatic tissues (MALTs) which are spread across the body. MALT functionalizes in maintaining the non-pathogenic conditions by upregulating the immune response at the site. Mucosal site present in the nasal cavity is named as Nasal-Associated Lymphatic tissue (NALT) located in the nasopharynx region. As the name suggest, NALT comprises various cells that are involved in immune function such as antigen-presenting cells like dendritic cells, macrophages, various helper T cells, and B cells [50,51] (Figure 3) Specialized epithelium cells called Microfold cells (M cells) that lack ciliated cells are the major site for the uptake of the foreign body by antigen-presenting cells. Induction of immune response is observed after the innate and adaptive immunity is triggered. Pattern recognition receptors present on epithelium cells signal to immune cells for antigen presentation. Production of cytokines like INF- β, TGF-β, IL-5, IL-6, and IL-10, IL-12, which act as chemotactic agents, occurs in this process that leads to cytotoxic effect and also recruit more antigen-presenting cells to the site of infection [52,53]. Moreover, DCs can also sample antigen across the epithelial barrier by transepithelial dendrite extension where the thin, flexible cytoplasmic protrusions are extended to the antigen and the antigen is uptaken [54]. Within the dendritic cells, antigen is processed, and the soluble antigen part is presented to the subsets of T helper cells through the major histocompatibility complex (MHC). B cell activation occurs when CD4^+^ T cells interact with B cells and on migration to local germinal center in NALT, differentiating into IgA-secreting plasma cells which bind to the polymeric immunoglobulin receptor (pIgR) on the basolateral side of epithelial cells. The IgA–pIgR complex is transported across epithelial cells to the apical (luminal) surface. At the surface, the pIgR is cleaved, and the secretory component remains bound to IgA, forming secretory IgA (sIgA). sIgA is released into nasal mucus where neutralization of antigen occurs [55].

#### 3.1.2. Limitations and Advancements in the Field

Singh et al. recently developed a combination intranasal vaccine for COVID-19 and influenza using AddaVax as the adjuvant, which elicited significantly higher mucosal IgA responses compared to the conventional intramuscular route [56]. Mucosal barrier is one of the reasons for liquid formulation retention. Mucociliary clearance removes substances from the nasal cavity in a short period of time thus limiting antigen exposure to immune cells. Moreover, nasal cavity has a limited volume of up to 250 uL, making high-dose delivery to a challenge [57,58]. Use of adjuvants is needed in live attenuated virus, nanoparticulate form, and sub-unit type vaccines to boost the immune response; however, the lack of marketed human mucosal adjuvants has been a challenge in the development of these formulations [59]. Safety concerns persist due to the proximity of the nasal cavity to the central nervous system, raising risks of neuroinflammation via olfactory transport, particularly with viral vectors or nanoparticles. Neurological manifestations associated with certain conditions can present in a variety of forms. Common symptoms include abnormal sensations such as tingling or numbness in the arms and legs, difficulty swallowing, and visual disturbances caused by inflammation of the optic nerve (optic neuritis). Individuals may also experience generalized muscle weakness, problems with coordination or walking, and, in some cases, seizures [57,60].

### 3.2. Oral Inhalation

Pulmonary delivery directly targets the infection portal, offering faster and more effective first-line defense. Moreover, it elicits both innate and adaptive immune responses at the site of infection. Delivering vaccines directly to the lungs mirrors the natural infection route for many respiratory pathogens. This stimulates strong local mucosal immune responses (particularly secretory IgA and tissue-resident memory T cells) in the respiratory tract, where the pathogen first invades, enhancing protection and reducing viral transmission.

Over the past decade, pulmonary vaccine delivery devices have advanced markedly to overcome the complex anatomical and physiological barriers of the respiratory tract, aiming to efficiently deposit antigens in the desired mucosal regions to induce robust local and systemic immunity. Device platforms used are nebulizers, dry powder inhalers, Metered Dose Inhalers (MDIs), and nanocarrier-enhanced delivery systems [50,61].

Nebulizers generate aerosols from liquid vaccine formulations, tailored to reach deep lung tissues, facilitating antigen uptake by dendritic cells and induction of mucosal immune responses. Advances focus on optimized particle size (1–5 µm) for maximal deposition in the lower respiratory tract and improved stability of fragile vaccines including mRNA and protein subunits during nebulization. Modern vibrating mesh nebulizers cause less shear stress, preserving antigen integrity better than traditional jet nebulizers. Several SARS-CoV-2 inhaled vaccines under clinical investigation utilize nebulizer-based delivery to promote mucosal sIgA and tissue-resident memory T cells in both upper and lower airways [62]. DPIs deliver vaccines in dry powder form, which increases stability, simplifies cold-chain requirements, and improves portability and ease of use, ideal for mass vaccination, especially in resource-limited settings. Advances in spray drying and particle engineering have enabled creation of virus-like particle powders and lipid nanoparticle-encapsulated RNA powders suitable for inhalation. Formulations focus on particle size control, flowability, and moisture resistance for consistent lung deposition. DPIs have been successfully employed in experimental pulmonary vaccines for influenza and SARS-CoV-2, showing enhanced mucosal IgA and systemic T cell responses compared to traditional injections; though, DPIs are less commonly used for vaccines so far. MDIs, on the other hand, offer precise dosing and rapid pulmonary delivery via pressurized propellants. Current research explores MDI formulation for nanoparticle-based vaccines, especially for improving delivery uniformity and patient compliance, but challenges related to vaccine stability and formulation in propellants still remain [63].

#### 3.2.1. Mechanism

Pulmonary delivery of vaccines occurs in the deep lungs especially in the lower airways of the respiratory region. Epithelial lining of the lymphatic tissue present in the lung’s bronchiole airway comprises immune cells in luminal and basal sides known as Bronchus-associated lymphoid tissue (BALT) [63]. Antigenic vaccine particles below 10 μm are transferred to these areas and are taken up by antigen-presenting cells like DCs by the mechanism of transcytosis or the M cells of the BALT. BALT comprises a densely packed network of lymphocytes, having follicular structures which are separated into B cell and T cell region [64]. B cells trigger the mucosal immune system by producing secretory IgA (sIgA) which is crucial for neutralizing pathogens at the mucosal surface, blocking attachment and entry into epithelial cells. Vaccination via the respiratory tract recruits and maintains CD4+ and CD8+ tissue-resident memory T cells in the airways and parenchyma. These cells act as sentinels, rapidly responding upon pathogen re-encounter by producing cytokines/chemokines, recruiting other immune cells, and providing immediate tissue protection [65]. Various surface proteins like integrins CD103 and/or CD49a and chemokine receptors CXCR3 and CXCR6 are also expressed, promoting retention and localization in the airway [66].

#### 3.2.2. Limitations of Inhalation Route

Vaccine antigens can be significantly degraded or denatured by environmental stresses during manufacturing, aerosolization, storage, and delivery, leading to reduced efficacy. Moreover, mucociliary clearance and enzymatic degradation destroys the antigen before it is captured by antigen-presenting cells (APCs), limiting the immune response. Particle size, shape, and formulation influence whether the vaccine reliably reaches the desired lung regions deep lung or upper airways. There is a need for precise engineering to ensure the vaccine reaches its target site without aggregating or being lost in the device or upper airways [67]. Device availability and standardization also remain challenges, especially for dry powder delivery in both preclinical and clinical settings. Achieving dose consistency can be harder with pulmonary delivery compared to injection. Patient factors (e.g., lung capacity, breathing patterns, age, health status) influence the amount of vaccine actually deposited and absorbed [68]. Maintaining the integrity and potency of inhaled vaccines through production, filling, and distribution is more challenging than with standard injectable vaccines [69]. In addition to this, direct delivery to the lungs can potentially cause unwanted pulmonary inflammation, bronchospasm, or irritation, especially if excipients are not optimized [61].

## 4. Microneedles Vaccine Delivery

Microneedle (MNS) are regarded as a transdermal medication delivery method that is minimally intrusive and painless to administer as it penetrates the stratum corneum. This is due to MNs’ ability to penetrate the skin without reaching the pain-producing nerve ends. MNs comprises a set of tiny needle-like projections, usually between 25 and 2000 μm in length, fabricated from materials like polymers, or alternative substances, and fixed onto a supporting base. These microneedles are engineered to reach skin regions rich in immune cells. Additionally, they can be independently administered, thereby fewer skilled healthcare workers are required, and there is no longer a chance of unintentional needle injuries or the requirement for proper needle disposal [70,71].

Ideal MN design requires reduced force while inserting and increased fracture force. Various factors influencing the efficacy of microneedles include the geometry of MN which impacts patient compliance, penetration effectiveness, and mechanical force. The sharpness influences the entry force of the MN while the tip diameter influences the depth of MN insertion. On the contrary, the sharpness decreases the mechanical robustness of the MN, causing it to crack. Studies show that sharp MN (less than 15 µm) are crucial for effective vaccines delivery. Since the blood vessels are in the dermis layer, the MN length needs to be long enough to reach the dermis to deliver the vaccine agent. In the case of shorter MN, there will be loss of therapeutic agent on the surface of the skin. However, very long MNs will induce the nerves which stimulate pain. To penetrate the stratum corneum and create microchannels, optimal force is required to accomplish targeted vaccine delivery. Studies showed that a force of around 20 mN was enough for optimal MN administration. Since manual applications may vary across individuals, an MN applicator is needed. Adding more needles in a closely spaced array enables increased drug loading, raising the quantity of microchannels that the vaccination can diffuse through. In this case, the total application force is distributed and shared across all the needles, which signifies that the insertion requires more pressure. This can be surpassed by the use of various lengths in the same array [71].

### 4.1. Mechanism of Action of Microneedles

One of the body’s important immune defense barriers is the integumentary system which offers both innate and acquired immunity. This structure consists of the most external stratum corneum, succeeded by the epidermis, dermis, and hypodermis. These layers comprise macrophages, keratinocytes, Langerhans cells (LCs), dendritic cells (DCs), and T cells. Keratinocytes recognize antigens through pattern-recognition receptors which lead to secretion of cytokines and chemokines. The LCs and DCs perform as antigen-presenting cells which collect the antigen and then present it to the T cells. Patrizia et al. showed that LCs were able to cross present antigen to CD8^+^ T cells and induce effector functions such as secretion of cytokines and cytotoxicity functions, playing a crucial role in epicutaneous vaccination strategies. Cross-presentation is important for producing immunity to viruses and inducing tolerance against self antigens [72]. When the LCs are stimulated by antigens, they mature, process the antigen, and go to specific lymph nodes that collect and filter lymph fluid coming from the skin area. Here, these cells activate both CD4^+^ T helper lymphocytes and CD8^+^ cytotoxic T lymphocytes. This mechanism concludes with B cell activation, which secretes antibodies and memory cells. This leads to robust humoral and cell-mediated immunity. In addition to MN, to use skin as entry site, a jet injector is a medical device that delivers vaccines using a high-pressure stream of liquid to penetrate the skin, eliminating the need for a needle. These devices, powered either by compressed gas or springs, have historically been used in large-scale immunization campaigns such as smallpox eradication and in military vaccination programs. The vaccine AFLURIA is approved for administration with a jet injector. AFLURIA protects against influenza A(H1N1), influenza A(H3N2), and one influenza B strain. When delivered by jet injector, AFLURIA is authorized for adults aged 18 to 64 years. The formulation intended for jet injector delivery is supplied in multi-dose vials that contain thimerosal.

### 4.2. Evolution of MN Technology in Vaccines

In 2002, the first study of solid MN vaccination was conducted by Mikszta et al. who used a micro-enhanced array to puncture the skin and achieve localized in vivo delivery of the vaccine marking the initial demonstration. This technology was found to decrease the number of vaccinations and generate more robust and consistent immune responses than needle-based injections. By using this technique, the vaccinations can efficiently enter the dermis, minimizing skin irritation and removing pain and discomfort [73]. In 2010, Fu-Shi Quan et al. successfully developed the first dissolving MN for influenza vaccination in vivo which generated superior protection compared to intramuscular route [74]. In 2015, For the first time, a dissolvable microneedle patch delivering a measles vaccine was evaluated in rhesus macaques and compared to the traditional subcutaneous injection method [75]. The use of MN vaccines has progressively expanded, encompassing immunizations such as influenza, rabies, and HPV. Researchers are exploring their application for COVID-19. The global toll of COVID-19 has reached millions of cases and fatalities. While vaccine candidates for SARS-CoV-2 are advancing swiftly, attaining herd immunity remains a key hurdle due to constraints like limited production, elevated costs, and dependence on refrigerated distribution. Microneedle-based vaccine delivery presents a compelling alternative to conventional injections, particularly for hard-to-reach groups, and holds promise for expanding immunization reach [76]. The first FDA approved MN for vaccine delivery was by Becton Dickinson in New Jersey, USA for BD Soluvia which is a microinjection system to ease and enhance the reliability of intradermal injections across people of different ages, genders, and races [77,78]. In 2009, The European Union authorized Intranza by Sanofi Pasteur’s, which was used to vaccinate adults against seasonal influenza, followed by approval of Fluzone by United States regulations in 2011, This led to the FDA’s publication of comprehensive regulatory guidelines for MNs in 2017.

### 4.3. Solid Microneedles

Innovative research discovered solid microneedles as the first model of transcutaneous delivery systems, including metallic, silicon-based, and non-biodegradable polymeric MNs. They function through ‘poke and patch’ approach, according to which microchannels are first formed, subsequently immunogenic agents are topically applied to the area of microchannel formation through which they pass through into the dermis (Figure 4). This technique initiates skin disruption which leads to localized microinjury, increasing the innate immune response and resulting in stronger adaptive immunity. The physical durability of solid MNs requires materials having strong mechanical integrity to penetrate the stratum corneum using ceramic, silicon, metallic materials. Polymeric solid MNs have been studied; however, they have shown lesser mechanical features. The detailed research of solid MNs as transcutaneous vaccination has shown positive preclinical findings; however, have some technical limitations such as needing two-phase administration regimen, possible accumulation of metal residue in skin layers, less ideal regulation on antigen release rate, and lack of uniform vaccine delivery. These limitations have led to engineering new MN prototypes, such as hollow and coated MNs [78].

### 4.4. Hollow Microneedles

The principle is based on pressure gradient flow or passive diffusion. This hollow configuration allows the delivery of considerably increased quantities of vaccine agents as they can hold few hundred microliters with improved flow in relation to other microneedle types. They can be made of metal, silicon, glass, or ceramics material. However, they are complicated to develop because of their framework and delicacy. In a study using ovalbumin as model antigen, in vitro skin permeation studies using Franz cell showed that hollow MNs had a stronger effect on skin permeation in contrast with solid MN and electroporation techniques. When the hollow MNs were utilized for the delivery of ovalbumin in vivo using mice model, they produced enhanced IgG immune response compared to subcutaneous injections. Hollow MNs immunization did not trigger symptoms of dermal infection or localized bleeding [79]. In a study utilizing hollow microneedles (MNs) for mumps and varicella vaccination in rats, intradermal delivery was achieved seamlessly, eliciting antibody responses for both vaccines that surpassed those induced by subcutaneous administration even with a reduced vaccine volume [80].

In a study by Zuo et al., rats were immunized with poliovirus vaccine combined with diphtheria–tetanus–acellular pertussis vaccine intradermally with hollow MN devices, namely MicronJet600, and the therapeutic effectiveness was analyzed and compared with intramuscular immunization. The humoral response of the combination vaccine delivered with MicronJet600 demonstrated that hollow MN approach had a dose-sparing effect which led to a more robust protection when the rats were infected with B pertussis. The hollow MN delivery proved to be a superior option compared to intramuscular immunization [81]. In a rat study, ovalbumin (a model antigen) and the TLR agonists imiquimod and monophosphoryl lipid A were encapsulated within PLGA nanoparticles and administered using a hollow microneedle array. This delivery method induced a significantly stronger IgG2a antibody response and increased the number of interferon (IFN)-γ secreting lymphocytes compared to intramuscular injection of antigen-loaded nanoparticles [82]. Yet, hollow MNs limit its use due to an elevated risk of needle tip blockage by tissue when inserted, obstructing vaccine flow. They also have high-cost production along with restriction of formulation of dry product and needle rupture.

### 4.5. Coated Microneedles

Coated MNs are solid MNs which are coated with a vaccine solution or dispersion by spray coating or dip coating, in which the MNs are immersed into the coating solution. Its usage limitation is due to restricted surface area for vaccine uptake. To overcome this, optimization of array dimensions by elevating needle density, or the application of gradual step-wise layer coating can be applied. In a study, Kim et al. optimized a formulation for MNs coated with influenza vaccine which were further administered transdermally to immunize mice. Coated MN immunization elicited strong systemic and functional antibody response and conferred full protection against fatal challenge infection equivalent to standard intramuscular injection. They concluded that antigen activity degradation during coating process can be avoided through optimized and strategic formulation [83]. In a study by Choi et al., they concluded that a combination of polyvinyl alcohol and trehalose is well-suited for coating for smallpox vaccination. Immunization with coated MNs generated neutralizing antibodies starting three weeks after vaccination which sustained over a twelve-week period. Cellular immunity was induced, as evidenced by markedly elevated numbers of IFN-γ-producing cells. The results showed that coated MNs offer an alternative delivery system for smallpox vaccination [84]. Park et al. identified inefficiencies in traditional microneedle (MN) coating methods, resulting in material waste. They employed electrospray deposition, a precise coating technique with near 100% efficiency and minimal material loss [85].

### 4.6. Dissolving Microneedles

Dissolving microneedles highlight improvements over other MNs by easier development and one-step administration. Dissolving MNs are fabricated using hydrophilic biodegradable polymers and function through pierce and release mechanism and gradually dissolve in the dermal layers post-penetration, leading to sustained release of vaccine followed by systemic absorption. In formulation of these systems, the liquid solutions of vaccines solidify in a controlled manner inside the molds. Reconstitution of therapeutic agents in a hyaluronic acid–trehalose gel, followed by spin-casting to fabricate dissolving microneedles, has been extensively investigated by Dsouza and colleagues for vaccines targeting Zika virus, SARS-CoV-2, gonorrhea, and influenza. The dissolving microneedle-based vaccination induced antibody responses that were equivalent to or exceeded those elicited by the conventional intramuscular route [85,86,87,88]. In a study by Kim et al., dissolving MNs were formulated for hepatitis B vaccine. The MNs pierced the skin efficiently and the base of MN fully dissolved within 10 min post-insertion, facilitating complete delivery of all MN tips into the skin. A single administration using dissolving MNs elicited antibody titers that matched or exceeded those achieved with two intramuscular injections [88].

### 4.7. Hydrogel/Swelling Microneedles

Hydrogel microneedles are made up using cross-linked hydrogels or high-swelling polymeric structure. They serve two roles: enabling vaccine delivery through regulated swelling process and facilitating diagnostic functions through selectively absorbing interstitial fluid from the skin. Vaccine agents can be loaded into hydrogel microneedles via two distinct approaches: First, vaccine incorporation at the base of MN and administration promotes hydrogel swelling by interstitial fluid uptake, creating pathways to the vaccine reservoir. Unlike dissolving MNs, these MNs allow removal with no residual polymer deposition in skin. The second approach incorporates vaccine and polymer mixing across the entire MN structure, achieving release of vaccine via swelling triggered by fluids. In a study by Courtenay et al., the hydrogel-forming MNs were compared with dissolving MNs using protein antigen ovalbumin. The mice vaccinated with dissolving MNs had significantly higher antibody titers than hydrogel MNs [89]. Further research is essential to evaluate swelling microneedle applications in vaccination.

## 5. Buccal Route of Administration

Among non-invasive immunizations, buccal administration, which targets the inner cheek mucosa, has emerged as a promising approach [90]. Buccal mucosa is highly vascularized and rich in lymphatic drainage, enabling efficient antigen absorption and systemic distribution while circumventing hepatic first-pass metabolism [91]. This contributes to improved antigen bioavailability and therapeutic efficacy. In addition to its immunological merits, buccal vaccination is non-invasive, painless, and potentially self-administrable, thus eliminating the need for sterile equipment or skilled personnel [29,92]. Furthermore, buccal vaccines may exhibit fewer gastrointestinal side effects compared to oral formulations, which are subjected to enzymatic degradation and pH variability in the digestive tract [93].

Notably, vaccines administered via the buccal route can stimulate the common mucosal immune system, leading to the production of both secretory IgA at mucosal surfaces and systemic IgG antibodies. This dual activation enhances immune protection not only at the site of administration but also at distal mucosal sites, such as the respiratory and gastrointestinal tracts. The trafficking of activated dendritic cells and IgA-secreting plasma blasts from the buccal mucosa to secondary lymphoid organs facilitates broad and durable immune responses [90,93].

### 5.1. Mechanistic Insights into Buccal Vaccine Delivery

The immunological mechanism underlying buccal vaccine delivery involves the efficient uptake and presentation of antigens by specialized mucosal immune cells [94]. Upon application to the buccal surface, antigens penetrate the epithelial barrier through transcellular (across cells) or paracellular (between cells) pathways, depending on their physicochemical properties. A key player in this process is the M cell (microfold cell), a specialized epithelial cell predominantly found in mucosa-associated lymphoid tissues (MALTs), including Peyer’s patches and, to a lesser extent, in the oral cavity. M cells actively transport luminal antigens across the epithelium into the underlying immune-rich lamina propria, where they are encountered by APCs such as dendritic cells and macrophages. These cells process and present antigens to T and B lymphocytes in local or draining lymph nodes, initiating both mucosal and systemic immune responses [94,95]. Strategies that specifically target M cells can further enhance vaccine efficacy due to their high transcytosis capacity and ability to bypass immune suppressant barriers within the mucosal environment (Figure 5).

The formulation of buccal vaccines also significantly influences their mechanism of action and immunogenicity. In conventional suspension forms, antigen absorption occurs via direct diffusion across the mucosal epithelium. Lipophilic molecules favor transcellular uptake, whereas hydrophilic, low-molecular-weight compounds traverse the paracellular route. Once internalized, antigens are processed by resident dendritic cells and transported to regional lymph nodes, initiating IgA- and IgG-mediated responses [93,96]. These antigens may also enter systemic circulation via the richly vascularized buccal tissue, enhancing immune activation. In contrast, nanoparticle-based vaccine formulations offer several advantages. Engineered for mucoadhesion and controlled release, these carriers protect antigens from enzymatic degradation and improve their retention time at the mucosal surface. Their particulate nature promotes uptake by M cells and APCs through phagocytosis or endocytosis, leading to more efficient antigen presentation and stronger immunological activation [92,94]. Nanoparticles can also be functionalized with ligands that target specific receptors on immune cells, enabling precision delivery and enhanced immune modulation. These features collectively result in improved immunogenicity, antigen stability, and durability of immune protection compared to conventional suspensions.

### 5.2. Current Research and Development

Early studies that contributed majorly to the advancements include Cui and Mumper (2002) [97], who developed mucoadhesive bilayer films for buccal genetic immunization in rabbits, marking a significant advance in needle-free vaccine delivery. The films featured a 3:1 blend of Noveon AA-1 and Eudragit S-100 forming a mucoadhesive layer and a pharmaceutical wax backing to ensure unidirectional antigen release and resistance to salivary washout. At ~109 µm thickness, the films demonstrated strong mucosal adhesion and mechanical stability, optimizing contact with the non-keratinized buccal epithelium rich in antigen-presenting cells. Post-loaded with 100 µg of plasmid DNA (CMV-β-gal) or protein antigen, the films released 60–80% of their payload within 2 h. Rabbits immunized buccally on days 0, 7, and 14 showed IgG titers comparable to subcutaneous controls and exhibited enhanced splenocyte proliferation, indicating cellular immune activation unique to mucosal DNA delivery. This study validated bilayer films as a stable, efficient, and immunogenic platform for buccal DNA vaccines.

A recent study [98] introduced a significant advancement in buccal vaccine delivery through the development of electro spun nanofibrous mucoadhesive films embedded with vaccine-loaded nanoparticles. These films were fabricated from biocompatible, mucoadhesive polymers such as polyvinyl alcohol (PVA) and chitosan, which enabled strong adhesion to the buccal mucosa while maintaining mechanical flexibility and structural integrity in the oral environment. Antigens were encapsulated in biodegradable nanoparticles, typically composed of poly(lactic-co-glycolic acid) (PLGA), to protect them from enzymatic degradation and salivary washout which are two of the key challenges associated with mucosal delivery. Ex vivo permeation studies using porcine buccal tissue demonstrated significant transmucosal penetration of the vaccine-loaded nanoparticles, while in vitro assays confirmed enhanced antigen uptake by human dendritic cells, indicating strong potential for initiating both mucosal and systemic immune responses. By directing antigen exposure to oral mucosa-associated lymphoid tissues (OALTs), the platform demonstrated the ability to stimulate both secretory IgA and systemic IgG production.

Esih et al. (2024) [99] presented a mucoadhesive film (MAF) platform for trans-buccal vaccine delivery, incorporating DNA, viral vector, and mRNA/LNP formulations targeting respiratory pathogens. Incorporation of SARS-CoV-2 mRNA/LNP constructs preserved structural integrity and enabled efficient penetration through antigen-presenting cell-rich buccal mucosa, inducing systemic IgG and mucosal IgA responses comparable to intramuscular injection, while uniquely promoting local IgA production critical for respiratory defense.

The study published in Vaccines (2024) [100] reported a significant advancement in buccal vaccine administration through the development of 3D-printed orally dissolving films (ODFs) loaded with microparticulate Zika virus vaccine formulations. Researchers employed a fully automated, precision-controlled bio fabrication process using a CELLINK INKREDIBLE+^®^ 3D bioprinter (Cellink, Gothenburg, Sweden) to generate reproducible, fast-dissolving polymer films designed to adhere to the moisture-rich, non-keratinized buccal mucosa for direct antigen delivery. The 3D-printed buccal films exhibited key attributes for mucosal vaccine delivery, including rapid disintegration (~3 min), consistent thickness (~0.25 µm), neutral pH (~7.1), and stability at room temperature for up to one year. SEM and DLS analyses confirmed uniform morphology and nanoparticle characteristics favorable for mucosal uptake. In vivo, buccal administration of adjuvanted Zika vaccine films induced strong IgG and IgA responses, along with enhanced CD4^+^ and CD8^+^ T cell activation. Anggraeni et al. reported a detailed investigation [101] into the influence of ovalbumin protein nanoparticle (OVA-NP) vaccine and alginate coating on macrophage uptake and immune activation, with implications for buccal vaccine delivery systems. The study focused on synthesizing ovalbumin-loaded protein nanoparticles of varying sizes using a desolvation method, subsequently modifying these nanoparticles with an alginate coating to improve stability and mucoadhesive properties relevant to mucosal administration. In vitro assays showed that smaller nanoparticles (~200 nm) had significantly higher uptake than larger ones (~400 nm). The hydrophilic, mucoadhesive alginate layer also prolonged mucosal retention by preventing rapid clearance. Confocal microscopy and flow cytometry confirmed efficient nanoparticle internalization and endosomal localization, essential for antigen processing and initiation of adaptive immunity. These findings highlight the need to precisely engineer nanoparticle size and surface chemistry to optimize buccal mucosal vaccine delivery.

Gala et al. reported another advancement in vaccine delivery through the buccal route [93]. The study detailed the physicochemical characterization and preclinical evaluation of a novel buccal measles vaccine formulated as microparticles (MPs) incorporated into ODFs. Spray-dried microparticles averaged 0.67 µm in size (0.5–0.9 µm), ideal for mucosal uptake and antigen presentation. These were uniformly embedded into mucoadhesive ODFs which rapidly dissolved in the buccal cavity, ensuring close contact with immune-active mucosa and uniform particle dispersion for reproducible dosing. Immunological assays showed robust T cell proliferation (>77% over six days) when dendritic cells were pulsed with vaccine-loaded microparticles. Confocal microscopy confirmed dendritic cell uptake and effective antigen presentation, thus confirming the formulation’s mechanical integrity, mucosal retention, and immune activation which highlight its technical potential.

## 6. Sublingual Vaccination

In sublingual delivery, the vaccines are administered beneath the tongue utilizing highly vascularized sublingual mucosa for antigen delivery [102,103]. The sublingual mucosa consists of three different anatomical layers which govern drug absorption characteristics. The outermost epithelial membrane consists of stratified squamous epithelial cells which serve a protective barrier function. Beneath this layer, we have the basement membrane, which replenishes the epithelium, followed by the lamina propria, a hydrated connective tissue layer consisting of collagen and elastic fibers, located below the basement membrane. The submucosa layer succeeds, which is a highly vascularized layer [104].

The sublingual region, with its non-keratinized stratified epithelium, exhibits greater permeability characteristics compared to other intraoral routes. This layer, providing more elastic and permeable tissue than the keratinized surfaces of the hard palate, facilitates rapid drug absorption [105]. The high vascularization combined with its profusion of capillaries beneath the epithelium enables fast drug diffusion into the venous circulation before entering the systemic circulation, ensuring efficient drug delivery [104]. The most crucial benefit is bypassing first-pass metabolism, as the drug avoids passing through the liver and enters systemic circulation directly [106]. This characteristic is particularly valuable when the drug is subject to high hepatic metabolism or degrades in a gastrointestinal environment. The rapid onset of action, typically occurring within minutes due to the rich blood supply and thin epithelial barrier, makes it an ideal route for acute therapeutic interventions that require prompt drug action, ensuring immediate relief for the patient. Additionally, nanovaccines, such as polymeric nanoparticles, liposomes, micelles, and nanofibers, offer benefits such as requiring lower doses, enhanced delivery effectiveness, targeted action, precise immune response activation, and suitable biocompatibility [107]. The nanofibers, which are made of mucoadhesive chitosan and polyethylene oxide via electrospinning, have been studied for their potential in sublingual drug delivery. Their strong mucoadhesive properties are due to intermolecular interactions between chitosan and mucin, specifically from bovine submaxillary glands, that enable these nanofibers to adhere to porcine sublingual mucosa ex vivo [108].

### 6.1. Mechanism of Sublingual Immunization

Sublingual vaccination utilizes the distinctive immune features of the oral mucosa to stimulate both systemic and mucosal immune responses (Figure 6). The sublingual mucosa contains specialized sublingual immune cell clusters (SLICs), which extend from the lamina propria to the epithelium, creating controlled microenvironments where T cells and CD11c^+^CD11b^+^ dendritic cells (DCs) reside in close proximity [109]. These specialized antigen-presenting cells can capture antigens, such as ovalbumin, within 30–60 min after sublingual administration, and the antigen itself can cross the epithelial barrier within 15–30 min [110]. Local and systemic immune responses are initiated after the activated dendritic cells migrate from the sublingual mucosa to the lymph nodes where naive T and B cells are primed. The result is the production of antigen-specific antibodies and cytotoxic T lymphocyte responses that can disseminate mucosal sites in the body [111]. Sublingual vaccines prevent the risk of antigen redirection to the nervous system and also provide mucosal and systemic immune protection [112].

### 6.2. Sublingual Tablets

Sublingual tablets are advanced pharmaceutical forms designed for placing under the tongue. They quickly dissolve to enable direct absorption of the drug through the highly vascularized sublingual mucosa. A clinical example of sublingual vaccine technology is MV140 (Uromune^®^, Inmunotek S.L., Madrid, Spain) which describes a pioneering example of a sublingual vaccine. In the case of MV140, a polyvalent inactivated whole cell bacterial sublingual tablet vaccine used to prevent recurrent urinary tract infections, human dendritic cells are activated by spleen tyrosine kinase (Syk) and MyD88-dependent signaling, leading to NF-kB and p38 MAPK activation that drives TH1, TH17, and IL-10, producing T cell differentiation and underpinning its clinical efficacy [113].

### 6.3. Patch-Based Systems

Mucoadhesive patch systems employ layer-by-layer (LbL) assembly of polysaccharides like chitosan and hyaluronic acid. This approach improves the retention time at the sublingual area and shields vaccine components from salivary dilution and enzymatic breakdown. The LbL mucoadhesive patches demonstrate sophisticated engineering approaches towards advancing sublingual vaccination [114].

Advanced patch designs feature Viscosan (VIS), a modified form of chitosan that breaks down more rapidly in saliva. However, research suggests that this quick degradation might reduce the effectiveness of nanoparticle delivery compared to traditional chitosan formulations. The findings emphasize the importance of adjusting the degradation rate of the patch to improve its contact time with mucosal tissues [114]. A novel hybrid approach that integrates microneedle technology with sublingual delivery methods is the Sublingual Dissolving Microneedle (SLDMN) patches. These devices feature micropillar compartments and are accompanied by 3D-printed applicators. Studies have shown that they effectively stimulate mucosal immunity through IgA production, reduce lung inflammation, and decrease cytokine levels in an SARS-CoV-2 vaccination model [115].

## 7. Vaginal Vaccines

Sexually transmitted diseases (STIs) such as herpes simplex virus (HSV), bacterial vaginosis (BV), and trichomoniasis continue to be a global contributor to illnesses, such as neurological and cardiovascular disease [116]. Many current vaccines, which are administered through traditional routes, often fail to induce a strong immune protection in genital tissue, a common entry for many pathogens [117]. Vaginal vaccine delivery offers an alternative due to its easy administration and strong mucosal immune response in the affected area.

### 7.1. Mechanism of Action

The vaginal mucosa provides a constant, multi-layered defense that includes physical, microbial, innate, and adaptive barriers. The vaginal epithelium comprises a stratified squamous epithelium that rests on a bed of stromal cells (Figure 7). The acidic antimicrobial matrix of this area helps to inhibit pathogen colonization. Within the epithelium, cervicovaginal Langerhans cells (cvLCs) use Langerin to capture and degrade viral particles, while CD14^−^ subepithelial dendritic cells (DCs) and CD14^+^ DCs/macrophages sample antigens via pattern-recognition receptors to traffic these antigens to local lymphoid tissues. The CD14^−^ populations support TH2-skewed/tolerant cytokines (IL-5, IL-13), while CD14^+^ APCs provide proinflammatory mediators to drive TH1 responses when a pathogen is sensed [118]. Underlying all of this is a resident network of resident lymphocytes and plasma cells that offer fast, site-specific immunity. CD8^+^ and CD4^+^ cells secrete IFN-γ and TNF upon further antigen exposure, which causes the epithelial cells to produce CXCL9/CXCL10. These chemokines will function in recruiting NK cells, DCs, and B and T Cells from circulation. The lamina propria plasma cells, which differentiate from activated B cells, predominantly secrete IgG, with IgA and IgM offering neutralization of pathogens on the mucosal surface. All of these work together in order to ensure protection from invasion [118].

### 7.2. History and Current Field of Vaginal Vaccines

It was long believed that the vagina was an ineffective route to deliver vaccines because of its weak immunogenic environment for adaptive immunity. However, in recent times, vaginal vaccination has been tested against numerous STIs and has shown its success in establishing populations of effector lymphocytes in the cervicovaginal mucosa [117]. In 2001, one of the earliest clinical trials on vaginal vaccine delivery was conducted, comparing the nasal and vaginal delivery of the model mucosal antigen toxin B subunit in 21 volunteers. Results showed the potential of vaginal vaccinations to induce strong immune responses. When given on days 10 and 24 of the menstrual cycle, cervical secretion yielded 58× more IgA and 16× more IgG, while nasal administration resulted in a 35× increase in overall vaginal IgA [119].

### 7.3. Limitations and Advancements

The fluctuating pH, high enzymatic activity, and consistent mucus turnover make antigens prone to rapid degradation. This limits immune activation and requires consistent or high doses to be effective [120]. Additionally, because of the menstrual cycle, hormones are constantly shifting, leading to more complications. Notably, estradiol affects the degree of vaginal immune response, mucus production, and thickening of the endometrium, leading to inconsistent antigen uptake [121]. Certain formulations also pose a risk because of the sensitivity of the vaginal lining, leading to irritation and discomfort [122]. This may make patients take extra caution or even deter them.

### 7.4. Mucoadhesive and Thermoresponsive Gels

Mucoadhesive and thermoresponsive gels were designed to address issues regarding the limited immune response of phosphate-buffered saline (PBS)-based vaccine vehicles. By using gels, challenges regarding clearance, degradation of antigen, and limited contact with mucosal surfaces were alleviated. Han et al. used poloxamers (Pol) as the primary polymer due to their high solubility, exceptional compatibility, and thermogelling properties. Poloxamers are inserted as a liquid and form a gel at the administration site. This protects the antigen from enzymatic degradation and allows it to be retained longer, thereby increasing immune activation. Mucoadhesive agents, such as polyethylene oxide (PEO) or polycarbophil (PC), can then be combined with the poloxamer-based polymer, allowing for improved vaginal residence and antigen–mucosal tissue interactions. The Pol/PEO or Pol/PC systems positively displayed enhanced mucosal and systemic immune responses to antigens such as HPV 16 L1 VLP and HbsAG [123]. To be more relevant to humans, Mehta et al. recently demonstrated in an ex vivo porcine vaginal model that Pluronic F127/F68 + sodium-alginate gel can retain a model protein for at least 8 h, as opposed to a rapid wash-out from a non-mucoadhesive control [10]. Thus, this porcine model, whose organs are closer in size and characteristically more similar to humans than mice, provides a strong bridge between mouse work and first-in-human gel-vaccine formulations [124]. In another important study, Park et al. provided evidence that HPV-16 L1 VLP + cholera toxin delivered in a Pol/PEO thermogelling vehicle elicits higher vaginal and salivary IgA titers, respectively, than the same dose delivered in PBS intravaginal or intramuscular routes. Additionally, after vaginal administration it was found that the Pol/PEO formulation elicited significantly greater systemic IgG compared to PBS controls. This shows that in situ mucosal gels can amplify both mucosal and systemic responses [125].

### 7.5. Mucus-Penetrating Vaccines

Mucus-penetrating nonviral gene vaccines might also alleviate the hindrances of standard vectors by passing through the layer of protective mucus hydrogel and the closely junctioned epithelial cells. Bi et al. created two < 200 nm lipopeptide-based carriers. The first system, DRRS, contained a charge reversal property to mimic viruses that are able to infect many different cell hosts. The second system, HA/RLS, had an HA coating which would allow it to directly target DCs. When DRLS meets the vagina’s acidic environment (pH 4), the DMMA cap is hydrolyzed, causing DRLS to lose its protective coat and exposing its positive charges. These positive charges enable it to stick to the epithelial cells, in turn allowing for the expression of HPV16 L1 antigens to DCs. Unlike unmodified RLS, both DRLS and HA/RLS gene complexes showed a rapid diffusion through mucus, about 40–50× faster than that of RLS. In vivo, acid-activated pseudo-DRLS, carrying EGFP or HPV16 L1 DNA, generated greater than double fluorescent intensity in excised reproductive tracts compared to PEI, and far more than HA/RLS, demonstrating widespread expression in the epithelium [126]. Most importantly, the DRLS complex carrying the HPV16 L1 gene was able to stimulate the expression of more antigens than the HA/RLS, meaning that there was a stronger cellular (0.9% IFN-γ^+^CD8^+^ and IL-4^+^CD4^+^ T cells) and mucosal (IgA) immunity. Additionally, the DRLS exhibited no evidence of epithelial damage, or alterations in liver and kidney function markers [126].

### 7.6. Intravaginal Rings

IVRs have long been used as an effective method for delivering drugs to the female reproductive tract, particularly for HIV prevention and contraception. These rings allow controlled drug diffusion, reducing the need for frequent dosing. IVRs are also coitally independent, further enhancing user convenience and integration into daily life [126].

One of the biggest challenges when it comes to IVRs is the application of this technology to the release of macromolecules (such as protein antigens). This issue is due to the thermal instability of proteins when subjected to the processing conditions needed in order to manufacture the rings. However, recent innovations have allowed rings to carry macromolecules. By embedding freeze-dried rods of hydroxypropyl methylcellulose (HPMC) containing recombinant protein antigens (and, as appropriate, TLR7/8 agonist adjuvant) in a silicone body cured at room temperature, protein integrity is maintained. These inserts consistently release >99% of antigen within 24 h in physiological buffer and simulated vaginal fluid, achieving a sustained burst of antigen at the mucosal surface [127].

#### On-Going Clinical Trials

Table 1 elaborates on the ongoing clinical trials for noninvasive immunization strategies. Intranasal vaccines are the most clinically advanced: BBV154 (iN-COVACC^®^) has completed Phase 3 evaluation as a single-dose COVID-19 booster and demonstrated robust systemic and mucosal immune responses, while influenza and COVID-19-based dNS1-RBD vaccines are undergoing Phase 1–3 trials in children and adults.

Oral plant-based platforms have also reached late-stage development; for instance, the Nicotiana-derived VLP vaccine Covifenz successfully completed Phase 3 trials against COVID-19, and MucoRice-CTB showed strong mucosal IgA induction in early clinical testing for cholera. Microneedle array patches are progressing rapidly, with Phase 1–3 trials for hepatitis B, measles–rubella, influenza, rotavirus, and polio reporting immunogenicity comparable or superior to traditional intramuscular vaccination, with added benefits of needle-free delivery, dose sparing, and improved acceptability. Other platforms remain in earlier stages: sublingual vaccines for COVID-19, influenza, and ETEC, as well as intravaginal vaccines for recurrent UTIs, vulvovaginal candidiasis, and HIV-1 have demonstrated promising induction of localized mucosal antibodies together with systemic responses in Phase 1–2 trials. Collectively, these data emphasize that non-invasive vaccines are not confined to conceptual or exploratory phases but are advancing toward clinical translation, with several already demonstrating measurable protection in humans.

## 8. Conclusions

The era of needle-free vaccination is rapidly approaching, driven by the convergence of immunology and patient-centric design. Non-invasive delivery routes spanning oral, intranasal, transdermal microneedles, buccal, and sublingual pathways are no longer conceptual novelties but viable platforms with growing preclinical and clinical evidence. Some challenges include mucosal surfaces being naturally inclined to foster tolerance, due to their constant exposure to non-harmful substances. This can result in weaker immune responses. Achieving sufficiently strong, long-lasting immunity often requires multiple doses and optimized antigen/adjuvant delivery strategies [157]. Needle-free vaccine technologies must address complex regulatory requirements for approval as combination products (device and drug). Looking ahead, future directions in noninvasive vaccine development are focused on overcoming barriers through a range of innovative approaches. Advances in delivery systems, such as nanoparticles, microparticles, and lipid-based carriers, are being refined to better protect antigens from degradation, enable targeted delivery to MALT, and provide controlled release for sustained immune activation. The development of next-generation mucosal adjuvants that are both potent and stable in the different mucosal regions is also a major research focus. Researchers are also working on host- and population-specific strategies, tailoring vaccine formulations to account for factors like malnutrition, microbiome differences, and age-related immune function to address variability in efficacy [158,159,160]. Improvements in manufacturing, such as advanced drying technologies and quality-by-design approaches, are enhancing the stability and scalability of noninvasive vaccines, reducing reliance on cold chains and making distribution more feasible worldwide [161,162]. As these technical and scientific challenges are addressed, noninvasive vaccines are poised to expand their reach to a broader range of diseases, including respiratory viruses, enteric pathogens, and even cancer. Finally, regulatory harmonization and global collaboration will be crucial for accelerating development, ensuring safety, and facilitating access to next-generation noninvasive vaccines. With continued interdisciplinary collaboration, non-invasive vaccines could redefine how we prevent infectious diseases by transforming vaccination from a clinical act to an everyday health behavior.

## Figures and Tables

**Figure 1 vaccines-13-00978-f001:**
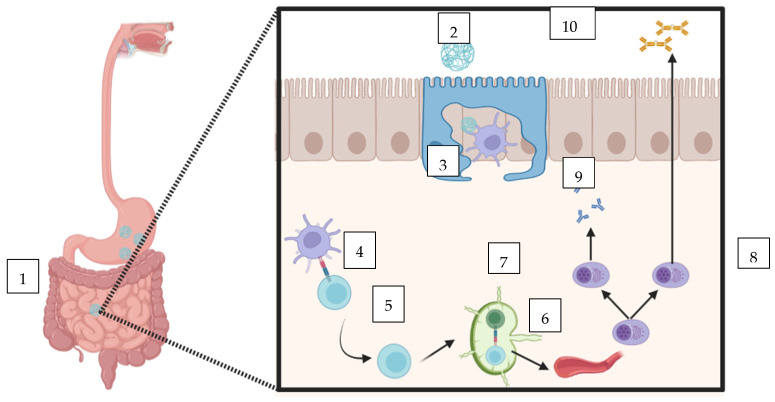
Vaccine antigen encapsulated in nanoparticle, shielding them from acidic pH, proteolytic enzymes, and bile salts in the GI tract, thereby preserving their biological activity. 2. Due to their small size, NPs can traverse mucosal barriers and diffuse through the mucus layer, reaching the underlying intestinal epithelium to be taken up by microfold (M) cells. 3. The antigen is released to be taken up by antigen-presenting cells. 4. Antigen presentation takes place leading to the activation of naïve T cells. 5. Activated T-cells go to lymph node, interact with B cells, and start to release plasma cells into the blood. 6. Plasma cells release IgG and secretory IgA, enhancing systemic and mucosal immunity. 7. IgG plasma cell. 8. IgA plasma cell. 9. IgG antibodies. 10. Secretory IgA antibodies.

**Figure 2 vaccines-13-00978-f002:**
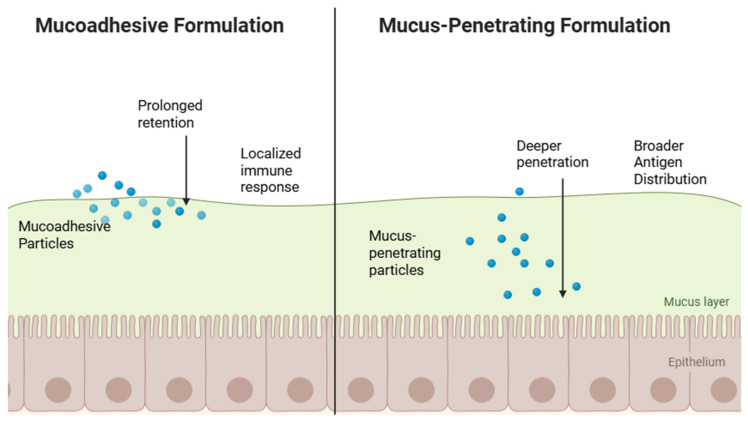
Mucoadhesive formulation vs. mucus-penetrating formulation.

**Figure 3 vaccines-13-00978-f003:**
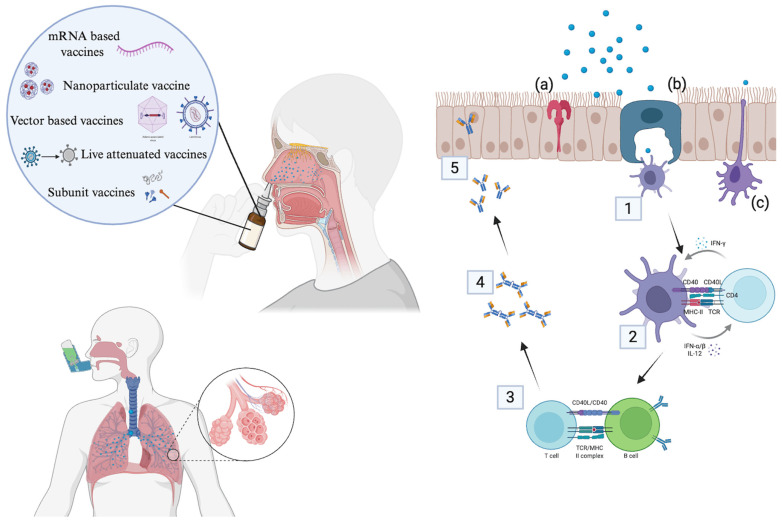
Illustration of inhalation route through intranasal and oral inhalation delivery of different platform of vaccine particles. Different vaccine like mRNA vaccines, nanoparticulate vaccines, vector-based vaccines, live attenuated vaccines, or subunit vaccines can be formulated separately and can be delivered using different devices into the nasal and the pulmonary tract targeting the mucosal associated lymphatic tissue. (**a**) Pattern recognition receptors in the epithelial layer recognize the antigen particles. (**b**) M cells in the epithelium transfer the antigen particles across the membrane. (**c**) Dendritic cells depend on stimulation strength to take up the antigen. (1) Antigen uptaken by DCs. (2) Antigen processed by DCs and presented through MHC-ll to CD4 helper T cells. (3) Naïve B cells activation interacts with T cells. (4) Dimeric IgA antibodies secretion. (5) Dimers bind to the polymeric immunoglobulin receptor (pIgR) and IgA secretory antibodies released in the nasal mucosa.

**Figure 4 vaccines-13-00978-f004:**
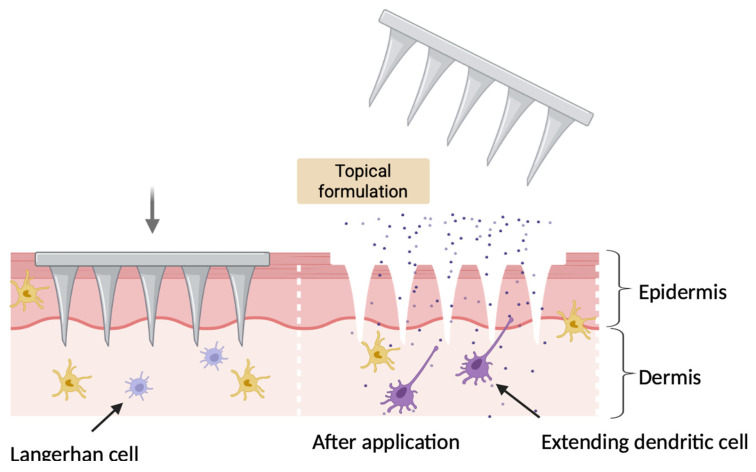
Representation of solid microneedle route. After application, the topical formulation is applied which is taken up by the dermal dendritic cells and Langerhans cells.

**Figure 5 vaccines-13-00978-f005:**
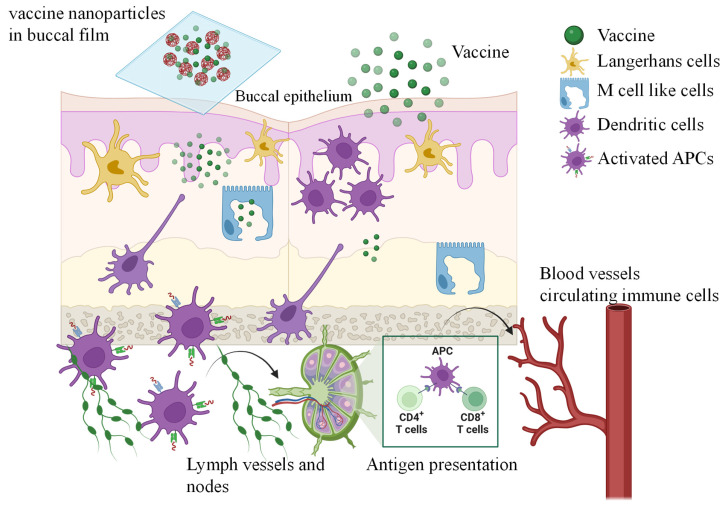
Representation of buccal administration route.

**Figure 6 vaccines-13-00978-f006:**
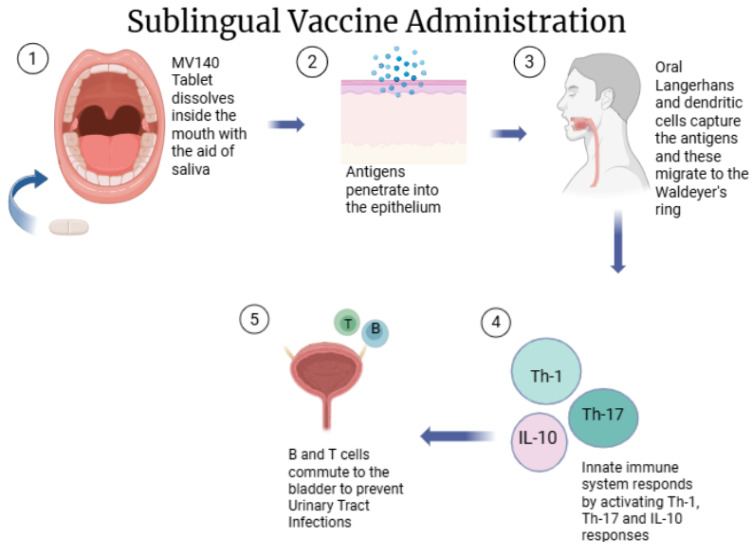
Representation of sublingual route.

**Figure 7 vaccines-13-00978-f007:**
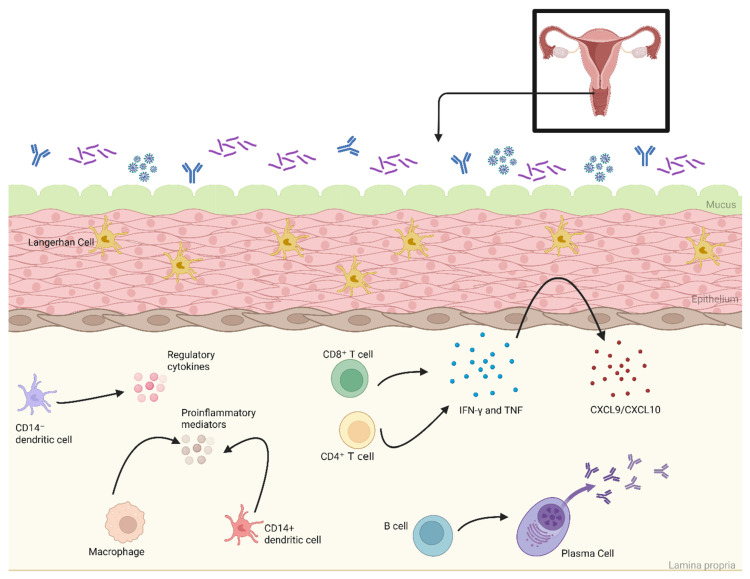
Depiction of the immune response in the lower female reproductive tract.

**Table 1 vaccines-13-00978-t001:** Clinical trials in non-invasive vaccine delivery.

Disease	Phase of Development	Name	Platform of Vaccine	Antigen and Adjuvant	
Intranasal Vaccines					
Influenza	Phase 1 completed [128]	FluGen + Fluzone High-Dose inactivated influenza vaccine	M2SR H3N2	Coadministration of the H3N2 M2SR vaccine with Fluzone HD	
COVID-19	Phase 2 Single-dose vaccine [129]	CVXGA Intranasal COVID-19 Vaccine in Adults	Parainfluenza virus 5 (PIV5) vector expressing SARS-CoV-2 spike	Omicron XBB.1.5 variant—Spike protein, chimeric virus formulated by inserting spike gene into the PIV5 genome	
COVID-19 + Influenza vaccine	Phase 3 Phase 1 trial—children aged 3–17 years [130]	Pneucolin^®^, ChiCTR2300068044	SARS-CoV-2 vaccine dNS1-RBD based on a live attenuated influenza virus vector	Cold-adapted influenza strain (CA4) without the non-structural protein 1 (NS1) as the genetic backbone, into which receptor-binding domain (RBD) gene from ancestral SARS-CoV-2 is inserted by gene reassortment	
COVID-19	Phase 3—completed [131]	BBV154 (iNCOVACC^®^) of 0.5 mL	Chimpanzee adenovirus-vectored, single-dose intranasal COVID-19 booster vaccine	Prefusion-stabilized SARS-CoV-2 spike	
COVID-19	Phase 2 trial [132]	NANOVAC (Intravacc)	Nanoparticulate peptide vaccine	HBcAg (hepatitis B core antigen) as an adjuvant and mucosal targeting carrier. soluble nanospheres containing synthetic mini-proteins from spike and other conserved coronavirus epitopes	
COVID-19	Phase 1 [63]	COVI-VACTM	Live-attenuated SARS-COV-2 synthetic viral vaccine	Attenuated through deletion of the furin cleavage site and introduction of 283 silent deoptimizing mutations that maintain viral amino acid sequence	
RSV	Phase 2 completed [133]	MV-012-968	recombinant, live attenuated RSV vaccine.		
RSV	Phase 1 in RSV-Seronegative Children	RSV/ΔNS2/Δ1313/I1314L	live attenuated, recombinant version of RSV strain A2	(1) A 523 nucleotide deletion of the NS2 gene and (2) a codon deletion in the L gene (Δ1313; deletion of S1313) plus the adjacent missense mutation I1314L that prevents the compensatory deattenuating mutation I1314T. The virus was generated from cDNA on World Health Organization Vero cells by reverse genetics	
Pertussis (whooping cough)	Phase 2b completed [134]	BPZE1 vaccine	Genetically modified, live attenuated strain of *Bordetella pertussis*	Modification by deletion of Dermonecrotic Toxin that eliminates local tissue damage, genetic inactivation of Pertussis Toxin (PT) to remove toxic effect of virus and gene expression reduced of Tracheal Cytotoxin that will prevent ciliary damage in airway epithelium	
**Oral Inhalation Vaccines**					
**Disease**	**Phase/Status**	**Product Name**	**Platform**	**Antigen and Adjuvant Preparation**	
COVID-19	Phase 2 (Canada) [135,136]	ChAd-triCoV/Mac (McMaster)	Viral vector (adenovirus, inhaled aerosol)	Triple-antigen: S1 spike, nucleocapsid, truncated RNA polymerase/non-adjuvanted	
COVID-19	Phase 2 (Canada) [137]	AeroVax (McMaster)	Viral vector (inhaled aerosol)	Adenovirus vector with 3 SARS-CoV-2 gene segments	
COVID-19	Phase 1–2 (China) [138]	Convidecia Air/Ad5-nCoV	Adenoviral vector (inhaled aerosol)	Adenovirus-vectored spike protein/non-adjuvanted	
**Microneedle vaccines**					
**Title of study**			**Type of Microneedle**		**Sponsor**
Phase 1: Hepatitis B Vaccine Delivered Trans-dermally by MAP [139]			Microneedle array patch (MAP)		International Vaccine Institute
Phase 2a, Establishing Immunogenicity and Safety of Needle-free Intradermal Delivery by Solid Micro Needle Skin Patch of mRNA SARS-CoV-2 Vaccine as a Revaccination Strategy in Healthy Volunteers [140]			Solid microneedle skin patch		Leiden University Medical Center
A Phase I/II, Double-blind, Randomized, Active-controlled, Age De-escalation Trial to Assess Safety and Immunogenicity of a Measles Rubella Vaccine (MRV) Microneedle Patch (MRV-MNP) in Adults, MRV-primed Toddlers, and MRV-naïve Infants [141].			Dissolving microneedle patch		Micron Biomedical, Inc.
A Phase 1 Study to Evaluate the Safety and Immunogenicity of CDC-9 Inactivated Rotavirus Vaccine for Intradermal Administration by Microneedle Patch in Healthy Adults [142].			Dissolving microneedle patch		Centers for Disease Control and Prevention
A Phase I Study of The Safety, Reactogenicity, Acceptability and Immunogenicity of Inactivated Influenza Vaccine Delivered either by Microneedle Patch or by Hypodermic Needle [143].			-		Mark Prausnitz
Phase 1 Evaluation of H1 Influenza Vaccine Delivered by MIMIX MAP [144].			MIMIX Microneedle Array Patch (MAP) System		Vaxess Technologies
A Pilot, Controlled, Comparative and Single Blinded Study to Evaluate the Safety and Immunogenicity of Low Dose Flu Vaccines Administered Intradermally Using Microneedle Injectors as Compared With Standard Dose Intramuscular Flu Vaccines as Reference [145].			-		NanoPass Technologies Ltd.
Phase III Clinical Trial to Assess the Immunogenicity of a Sequential Dose of Fractional Inactivated Polio Vaccine (f-IPV) and Oral Polio Vaccine (OPV) [146].			MicroJet 600 microneedle		Centers for Disease Control and Prevention
**Sublingual Vaccination**					
**Disease**	**Phase**	**Name**	**Platform Type**	**Antigen/Adjuvant**	
COVID-19	Phase 1 [147]	hAd5 T cell vaccine	Adenoviral Vector	Spike + Nucleocapsid proteins	
Influenza (QIV)	Phase 1 [148]	Tablet (NIBRG-14)	Inactivated virus tablet	H5N1, inulin glass	
ETEC (traveler’s diarrhea)	Phase 1 [149]	CfaEB-dmLT vaccine	Subnit + vaccine	CfaEB/CfaE-LTB, dmLT (mutant LT)	
**Oral vaccination**					
**Disease**	**Phase of Development**	**Name/Developer**	**Vaccine Platform**	**Plant Used**	**Antigen and Adjuvant**
Cholera	Phase I/II [150]	MucoRice-CTB, Seed-based vaccines	Subunit (CTB, VLP)	Rice, tobacco	CTB (cholera toxin B); none/various
Norwalk virus	Early Clinical [151]	Various	VLP	Potato, tobacco	Norwalk virus capsid protein; none
Rotavirus	Phase I [42]	Maize-based bivalent vaccine, Ro-VLP	Subunit (VP6, NSP4, VLP)	Maize, potato, alfalfa	VP6, NSP4, Ro-VLP; LTB (*E. coli* heat-labile toxin B)
COVID-19	Phase III, Registered [152]	Covifenz (Medicago/GSK), Baiya Phytopharm	Virus-like Particle (VLP)	Nicotiana benthamiana	Spike (S) protein; AS03/CpG1018 adjuvant
Enterotoxigenic E. coli	Phase I (human), Preclinical (animal) [153]	-	Subunit	Potato, maize	LT-B (heat-labile toxin B subunit); none
**Intravaginal Vaccines**					
**Disease**	**Phase**	**Name of product**	**Platform**	**Antigen and Adjuvant**	
Recurrent Vulvovaginal Candidiasis	Phase I (completed) [154]	PEV7	Virosome-formulated subunit vaccine delivered either via intravaginal capsule (PEV7C) or IM injection (PEV7B)	Sap2 protein (Candida albicans)/non-adjuvanted	
Recurrent UTIs	Phase II [155]	Urovac™	Vaginal suppository	10 strains of uropathogenic bacteria that were heat-killed/non-adjuvant	
HIV-1	Phase I [156]	CN54 gp140	Recombinant HIV-1 enveloped protein formulated in a Carbopol gel, delivered via applicator	HIV-1 (CN54 gp140)/non-adjuvanted	

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
