# Peer review of "Recent Advancements in Non-Invasive Vaccination Strategies"

_vaccines, 2025, doi:10.3390/vaccines13090978_

Round 1
Reviewer 1 Report
Comments and Suggestions for Authors
The microneedles section limited mainly to preclinical data and is missing discussion and references to the many clinical trials that have been conducted with hollow, coated, and dissolving microneedles with vaccines such as influenza, measles-rubella, JE, and COVID-19.
The description of hollow microneedles is inaccurate and confusing. The vaccine is not "sited inside the pointed ends of the needles". It is generally in a syringe or reservoir and travels through the hollow microneedles into the skin.
It is incorrect to say that microneedle vaccines do not require cold chain. It varies by vaccine antigen, but microneedles delivering dry formulations generally must be stored in the cold chain for most of their shelf life, and some are stable enough to be kept at room temperature for a period of days or weeks just before use. Hollow microneedles delivering liquid injections have the same cold chain storage as any other vaccine.
The section on ocular drug delivery seems irrelevant to the subject of the article.
I don't think a vaginal jet injector would be considered "non-invasive" to the recipient. Why are skin jet injectors not included?
Comments on the Quality of English LanguageThis manuscript badly needs thorough proofreading and editing, there are many errors of grammar and capitalization throughout.
Author Response
The microneedles section limited mainly to preclinical data and is missing discussion and references to the many clinical trials that have been conducted with hollow, coated, and dissolving microneedles with vaccines such as influenza, measles-rubella, JE, and COVID-19.
Response to reviewer comment
We thank the reviewer for bringing up this important point. The clinical trials can be found in table 2, towards the end of the manuscript. In Table 2 we have included and cited microneedle vaccination clinical trials on-going for Hepatitis B, Measles Rubella Vaccine (MRV) Microneedle Patch, Inactivated Rotavirus Vaccine, Inactivated Influenza Vaccine H1 Influenza Vaccine and Inactivated Polio Vaccine Low Dose Flu Vaccines.
The description of hollow microneedles is inaccurate and confusing. The vaccine is not "sited inside the pointed ends of the needles". It is generally in a syringe or reservoir and travels through the hollow microneedles into the skin.
Response to reviewer comment
We thank the reviewer for pointing it out. We have removed the description of ‘sited inside the pointed ends of the needles’ from the updated manuscript.
It is incorrect to say that microneedle vaccines do not require cold chain. It varies by vaccine antigen, but microneedles delivering dry formulations generally must be stored in the cold chain for most of their shelf life, and some are stable enough to be kept at room temperature for a period of days or weeks just before use. Hollow microneedles delivering liquid injections have the same cold chain storage as any other vaccine.
Response to reviewer comment
We thank the reviewer for their suggestion. We have removed lines 334-337 from the manuscript which mentioned cold chain storage related to microneedles.
The section on ocular drug delivery seems irrelevant to the subject of the article.
Response to reviewer comment
We thank the reviewer for their valuable comment for the improvement of the manuscript. We have removed lines 618-658 of the previous manuscript which was the section of ocular vaccination from the updated manuscript.
I don't think a vaginal jet injector would be considered "non-invasive" to the recipient. Why are skin jet injectors not included?
Response to reviewer comment
We thank the reviewer for raising this important point. We agree completely and have removed lines 751-767 of the manuscript which was the section of the vaginal jet injector. We have added a section on skin jet injectors on lines 382-389: “ In addition to MN to use skin as entry site, a jet injector is a medical device that delivers vaccines using a high-pressure stream of liquid to penetrate the skin, eliminating the need for a needle. These devices, powered either by compressed gas or springs, have historically been used in large-scale immunization campaigns, such as smallpox eradication, and in military vaccination programs.The vaccine AFLURIA is approved for administration with a jet injector. AFLURIA protects against influenza A(H1N1), influenza A(H3N2), and one influenza B strain. When delivered by jet injector, AFLURIA is authorized for adults aged 18 to 64 years. The formulation intended for jet injector delivery is supplied in multi-dose vials that contain thimerosal.”
Reviewer 2 Report
Comments and Suggestions for Authors
Gulani et al in the manuscript “Recent Advancements in Non-Invasive Vaccination Strategies” comprehensively reviews recent advancements in non-invasive vaccination strategies, highlighting their potential to overcome limitations of needle-based delivery by inducing robust mucosal and systemic immunity. While the manuscript thoroughly covers mechanisms, formulations, and clinical advancements, its impact is somewhat undermined by several structural and content-related issues that affect its overall rigor and readability. Below are several points the authors may consider enhancing:
- The introduction section is disproportionately lengthy and revisits well-established general knowledge regarding vaccines and cold-chain challenges, which delays engagement with the core focus of the review. Condensing this section would provide a more efficient and focused lead-in to the main topic of non-invasive vaccination strategies.
- The review exhibits an imbalance in depth across the various vaccine delivery routes. While oral and microneedle-based vaccines are discussed in extensive detail, encompassing multiple sub-formulations and mechanistic insights, other routes such as ocular and vaginal vaccination receive comparatively superficial treatment. Addressing this disparity would improve the comprehensiveness of the review and offer readers a more balanced perspective on the entire non-invasive vaccine landscape.
- The conclusion is notably brief and does not adequately synthesize the key findings presented throughout the manuscript. Rather than simply heralding the arrival of needle-free vaccination, a more robust conclusion should summarize the most promising platforms, integrate challenges highlighted across different delivery routes, and outline clear future directions for research and clinical translation.
- Some statements require more precise phrasing to accurately reflect current evidence. For example, the claim that “…nasal delivery of vaccine also lacks to produce systemic IgG antibodies and memory markers…” may be overly absolute, as numerous studies have demonstrated that intranasal vaccines can indeed induce systemic immune responses, including IgG production. Clarifying such points would strengthen the manuscript's scientific accuracy.
Author Response
- The introduction section is disproportionately lengthy and revisits well-established general knowledge regarding vaccines and cold-chain challenges, which delays engagement with the core focus of the review. Condensing this section would provide a more efficient and focused lead-in to the main topic of non-invasive vaccination strategies.
Response to reviewer comment
We thank the reviewer for their valuable suggestion to improve the manuscript. We acknowledge that the introduction previously included general information on vaccines and detailed discussion on cold chain challenges. Accordingly, we have removed lines 27–31 and 42–46 from the introduction and condensed the section.
- The review exhibits an imbalance in depth across the various vaccine delivery routes. While oral and microneedle-based vaccines are discussed in extensive detail, encompassing multiple sub-formulations and mechanistic insights, other routes such as ocular and vaginal vaccination receive comparatively superficial treatment. Addressing this disparity would improve the comprehensiveness of the review and offer readers a more balanced perspective on the entire non-invasive vaccine landscape.
Response to reviewer comment
We thank the reviewer for this observation. Considerable progress has been made in oral, microneedle, buccal, and intranasal vaccinations, which is why these routes were discussed in greater detail. We have removed the ocular section. With regard to vaginal delivery, limited advancements have been reported to date; therefore, only a small number of relevant studies were included.
- The conclusion is notably brief and does not adequately synthesize the key findings presented throughout the manuscript. Rather than simply heralding the arrival of needle-free vaccination, a more robust conclusion should summarize the most promising platforms, integrate challenges highlighted across different delivery routes, and outline clear future directions for research and clinical translation.
Response to reviewer comment
We agree with the reviewer and thank them for raising this crucial detail. We have added lines 764-782 to the conclusion part of the updated manuscript. This highlights the limitations and future directions to noninvasive vaccination. We have added the following paragraph:
‘Some challenges include- mucosal surfaces are naturally inclined to foster tolerance, due to their constant exposure to non-harmful substances. This can result in weaker immune responses. Achieving sufficiently strong, long-lasting immunity often requires multiple doses and optimized antigen/adjuvant delivery strategies [152].Needle-free vaccine technologies must address complex regulatory requirements for approval as combination products (device and drug). Looking ahead, future directions in noninvasive vaccine development are focused on overcoming barriers through a range of innovative approaches. Advances in delivery systems, such as nanoparticles, microparticles, and lipid-based carriers, are being refined to better protect antigens from degradation, enable targeted delivery to MALT and provide controlled release for sustained immune activation. The development of next-generation mucosal adjuvants that are both potent and stable in the different mucosal regions is also a major research focus. Researchers are also working on host- and population-specific strategies, tailoring vaccine formulations to account for factors like malnutrition, microbiome differences, and age-related immune function to address variability in efficacy[153–156]. Improvements in manufacturing, such as advanced drying technologies and quality-by-design approaches, are enhancing the stability and scalability of non invasive vaccines, reducing reliance on cold chains and making distribution more feasible worldwide[157,158]. As these technical and scientific challenges are addressed, non invasive vaccines are poised to expand their reach to a broader range of diseases, including respiratory viruses, enteric pathogens, and even cancer. Finally, regulatory harmonization and global collaboration will be crucial for accelerating development, ensuring safety, and facilitating access to next-generation noninvasive vaccines.’
- Some statements require more precise phrasing to accurately reflect current evidence. For example, the claim that “…nasal delivery of vaccine also lacks to produce systemic IgG antibodies and memory markers…” may be overly absolute, as numerous studies have demonstrated that intranasal vaccines can indeed induce systemic immune responses, including IgG production. Clarifying such points would strengthen the manuscript's scientific accuracy.
Response to reviewer comment
We thank the reviewer for pointing this out, We have removed the following sentence from the updated manuscript. “Nasal delivery of vaccine also lacks to produce systemic IgG antibodies and memory markers which are the one of indicator of robust immune response, thus provides localized immune response.”
Reviewer 3 Report
Comments and Suggestions for Authors
This review article ("Recent Advancements in Non-Invasive Vaccination Strategies") by Gulani et al. discusses the recent advancements in non-invasive vaccination strategies, especially oral, intranasal, mi-croneedle, buccal, sublingual, ocular, and vaginal vaccinations and highlights their underlying immunological mechanisms, formulation strategies in preclinical studies, examples of marketed products, and ongoing clinical trials. The review is very well written, quite extensive, and informative as well. I, therefore, recommend it for publication in the Vaccines journal.
Author Response
This review article ("Recent Advancements in Non-Invasive Vaccination Strategies") by Gulani et al. discusses the recent advancements in non-invasive vaccination strategies, especially oral, intranasal, mi-croneedle, buccal, sublingual, ocular, and vaginal vaccinations and highlights their underlying immunological mechanisms, formulation strategies in preclinical studies, examples of marketed products, and ongoing clinical trials. The review is very well written, quite extensive, and informative as well. I, therefore, recommend it for publication in the Vaccines journal.
Response to reviewer comment-
We thank the reviewer for their kind comment.
Reviewer 4 Report
Comments and Suggestions for Authors
The manuscript presents a review of advances and perspectives in the development of vaccines administered by alternative routes, including oral, intranasal, and other unconventional routes. It explores immunological mechanisms, formulations undergoing preclinical evaluation, currently available products, and ongoing clinical trials. This is a relevant and current topic, sparking interest due to its technological innovation and potential impact on public health. The title adequately reflects the article's purpose; however, some central aspects could be addressed in greater depth to enrich the scientific contribution.
First, the objectives address several points superficially. Issues such as the immunological mechanisms involved, the development stages of the platforms, the degree of protection conferred, and concrete examples of vaccines under study or in use deserve greater detail. Furthermore, I missed the inclusion and discussion of already established and widely used vaccines, such as DTP and other combinations, which are essential in both developed and developing countries and could serve as a comparative reference.
In the introduction, specifically on page 1, the authors cite the need for new vaccine delivery and development technologies in developed countries. However, it is important to emphasize that this need is global. Several developing countries are producers of established vaccines and have a growing capacity for innovation. Therefore, it is recommended that the text be reworded to avoid dichotomies between "developed" and "undeveloped" countries, adopting a universal approach, since the challenges and advances in vaccination are of interest and affect the entire world.
In section 2, where the authors subdivide the different types of administration/action in the context of the so-called New Wave of Oral Vaccine Research, it would be very useful to add concrete examples. The reader would benefit from a clearer presentation of which diseases or illnesses are being targeted by these strategies, accompanied by information on the stages of development (preclinical, phase I–III clinical trials, or already approved). This approach would make the analysis more robust and facilitate understanding of the current and future landscape of these technologies.
In summary, the manuscript addresses a highly relevant topic. It has potential for impact, but it lacks greater depth in its discussion of immunological mechanisms, examples of both classic and new vaccines, and the global context of the need for innovation. A more detailed review of these aspects would certainly strengthen the article, making it more comprehensive and useful for the scientific community and professionals involved in vaccine research and development.
Author Response
- First, the objectives address several points superficially. Issues such as the immunological mechanisms involved, the development stages of the platforms, the degree of protection conferred, and concrete examples of vaccines under study or in use deserve greater detail. Furthermore, I missed the inclusion and discussion of already established and widely used vaccines, such as DTP and other combinations, which are essential in both developed and developing countries and could serve as a comparative reference.
Response to reviewer comment-
We thank the reviewer for this valuable suggestion. Our focus in this review was primarily on emerging non-invasive vaccine platforms, and for this reason we did not go into detail on well-established injectable vaccines such as DTP and its combinations. However, we agree that including a brief discussion of such widely used vaccines will provide important context and serve as a useful point of comparison when highlighting the advantages and limitations of non-invasive approaches. We have therefore added a section referencing DTP and other conventional combination vaccines to underscore their historical impact and global relevance.
Lines 40-45: Conventional vaccines such as diphtheria–tetanus–pertussis (DTP) and their combination formulations (e.g., DTP-HepB-Hib, DTP-HepB-Hib-IPV) represent some of the most widely used and successful immunization strategies globally. These injectable vaccines have played a pivotal role in reducing childhood morbidity and mortality in both developed and developing countries, and they continue to serve as a cornerstone of routine immunization programs recommended by the WHO and national agencies. Despite their effectiveness, these approaches face practical challenges that can hinder their widespread use.
We acknowledge that greater detail for on-going clinical trials is required. We have added Lines 743-756-
Table 2 elaborates on the on-going clinical trials for non invasive immunization strategies. Intranasal vaccines are the most clinically advanced: BBV154 (iN-COVACC®) has completed Phase 3 evaluation as a single-dose COVID-19 booster and demonstrated robust systemic and mucosal immune responses, while influenza and COVID-19-based dNS1-RBD vaccines are undergoing Phase 1–3 trials in children and adults. Oral plant-based platforms have also reached late-stage development; for instance, the Nicotiana-derived VLP vaccine Covifenz successfully completed Phase 3 trials against COVID-19, and MucoRice-CTB showed strong mucosal IgA induction in early clinical testing for cholera. Microneedle array patches are progressing rapidly, with Phase 1–3 trials for hepatitis B, measles-rubella, influenza, rotavirus, and polio reporting immunogenicity comparable or superior to traditional intramuscular vaccination, with added benefits of needle-free delivery, dose sparing, and improved acceptability. Other platforms remain in earlier stages: sublingual vaccines for COVID-19, influenza, and ETEC, as well as intravaginal vaccines for recurrent UTIs, vulvovaginal candidiasis, and HIV-1, have demonstrated promising induction of localized mucosal antibodies together with systemic responses in Phase 1–2 trials. Collectively, these data emphasize that non-invasive vaccines are not confined to conceptual or exploratory phases but are advancing toward clinical translation, with several already demonstrating measurable protection in humans.
- In the introduction, specifically on page 1, the authors cite the need for new vaccine delivery and development technologies in developed countries. However, it is important to emphasize that this need is global. Several developing countries are producers of established vaccines and have a growing capacity for innovation. Therefore, it is recommended that the text be reworded to avoid dichotomies between "developed" and "undeveloped" countries, adopting a universal approach, since the challenges and advances in vaccination are of interest and affect the entire world.
Response to reviewer comment-
We appreciate the reviewer’s suggestion towards improvement of the manuscript. We have removed all the text relating developed and undeveloped countries. This can be found on line 27-line 36 of the updated manuscript.
- In section 2, where the authors subdivide the different types of administration/action in the context of the so-called New Wave of Oral Vaccine Research, it would be very useful to add concrete examples. The reader would benefit from a clearer presentation of which diseases or illnesses are being targeted by these strategies, accompanied by information on the stages of development (preclinical, phase I–III clinical trials, or already approved). This approach would make the analysis more robust and facilitate understanding of the current and future landscape of these technologies.
Response to reviewer comment-
We appreciate the reviewer’s valuable. All ongoing clinical trials for oral vaccines are included in Table 2, covering Cholera, Rotavirus, COVID-19, and Enterotoxigenic E. coli, along with details on the trial phase, vaccine platform, antigen, and adjuvant used.
- In summary, the manuscript addresses a highly relevant topic. It has potential for impact, but it lacks greater depth in its discussion of immunological mechanisms, examples of both classic and new vaccines, and the global context of the need for innovation. A more detailed review of these aspects would certainly strengthen the article, making it more comprehensive and useful for the scientific community and professionals involved in vaccine research and development.
Response to reviewer comment-
We acknowledge the reviewer’s comment towards improvement of the manuscript. We have added the following lines to the updated manuscript.
Lines 536-545- Early studies that contributed majorly to the advancements include Cui and Mumper (2002) [93] developed mucoadhesive bilayer films for buccal genetic immunization in rabbits, marking a significant advance in needle-free vaccine delivery. The films featured a 3:1 blend of Noveon AA-1 and Eudragit S-100 forming a mucoadhesive layer, and a pharmaceutical wax backing to ensure unidirectional antigen release and resistance to salivary washout. At ~109 µm thickness, the films demonstrated strong mucosal adhesion and mechanical stability, optimizing contact with the non-keratinized buccal epithelium rich in antigen-presenting cells. Post-loaded with 100 µg of plasmid DNA (CMV-β-gal) or protein antigen, the films released 60–80% of their payload within 2 hours. Rabbits immunized buccally on days 0, 7, and 14 showed IgG titers comparable to subcutaneous controls and exhibited enhanced splenocyte proliferation, indicating cellular immune activation unique to mucosal DNA delivery. This study validated bilayer films as a stable, efficient, and immunogenic platform for buccal DNA vaccines.
Lines 120-130- Bacillus subtilis spores are safe, spore-forming, and scalable vectors that can withstand the harsh gastrointestinal (GI) environment, present antigens, and trigger both innate and adaptive immune responses. They have been explored for vaccines against tetanus, rabies, and rotavirus, though they require repeated dosing and pose regulatory and genetic safety challenges. Lactic acid bacteria (LAB), such as L. plantarum and L. lactis, are acid-resistant and can target Peyer’s patches, where they colonize the gut, present antigens to mucosal immune cells, and induce secretory IgA and systemic IgG responses. They have been investigated for vaccines against enterotoxigenic E. coli, COVID-19, and Lyme disease, but face issues of variable immunogenicity, production difficulties, and unpredictable host immune responses. Yeast-based vectors, including S. cerevisiae and P. pastoris, have robust cell walls and are highly engineerable, enabling them to protect antigens, promote immune uptake, and induce both humoral and cellular immunity. They have been applied in hepatitis B and influenza vaccines and evaluated in various preclinical and clinical studies, but achieving consistent antigen expression and meeting regulatory requirements remain challenges.
Lines 131-145:Bacillus subtilis–based vaccines have been evaluated in several preclinical studies. For tetanus[18], recombinant B. subtilis spores or vegetative cells expressing the tetanus toxin fragment C (TTFC), sometimes combined with E. coli mLT as an adjuvant, have been tested in mice and pigs. Oral rabies vaccine candidates [19] such as CotG-E-G and CotG-C-G, displaying rabies virus glycoprotein, showed promise in mice. Similarly [20], recombinant B. subtilis expressing rotavirus proteins (VP8* or VP6), often delivered as spores or vegetative cells and sometimes paired with mucosal adjuvants like cholera toxin or E. coli mLT, were evaluated in mice. Lactic acid bacteria (LAB) have also been explored [21], Lactococcus lactis NZ3900 and Lactobacillus casei were engineered against enterotoxigenic E. coli (ETEC), expressing mutant heat-labile toxin subunits (LTA, LTB) and F4 fimbriae (FaeG), and tested in piglets and mice. LAB strains including L. lactis IL1403, NZ3900, and Lactiplantibacillus plantarum have been used to deliver SARS-CoV-2 spike or receptor-binding domain (RBD) antigens in mouse COVID-19 models [22]. For Lyme disease[23], L. plantarum was engineered to present Borrelia burgdorferi surface antigens such as OspA and studied in mice. Yeast-based vectors have reached both preclinical and clinical testing: recombinant Saccharomyces cerevisiae expressing hepatitis B surface antigen (HBsAg) [24] underlies licensed vaccines like Engerix-B and Recombivax HB, as well as oral prototypes; and yeast-displayed influenza vaccines[25], including H5N1 hemagglutinin and recombinant virus-like particle platforms, have been investigated preclinically.
Lines 208-212 In parallel, bacterial-like particles (BLPs) derived from lactic acid bacteria have been studied as oral vaccine platforms. For enteric bacterial infections[33], candidates such as COB17 and LM@COB17 used lipid-coated BLPs conjugated with model protein antigens, tested in preclinical settings without additional adjuvants. Similarly, BLPs have been engineered to display HIV-1 gp120 Env trimer antigens, representing a preclinical oral vaccine approach against HIV-1[44].
Round 2
Reviewer 1 Report
Comments and Suggestions for Authors
None
Reviewer 2 Report
Comments and Suggestions for Authors
I have no futher questions.
Reviewer 4 Report
Comments and Suggestions for Authors The authors have significantly improved the manuscript, and we now consider it suitable for publication.